# Headaches and Dizziness as Disabling, Persistent Symptoms in Patients with Long COVID–A National Multicentre Study

**DOI:** 10.3390/jcm11195904

**Published:** 2022-10-06

**Authors:** Mª Pilar Rodríguez-Pérez, Patricia Sánchez-Herrera-Baeza, Pilar Rodríguez-Ledo, Sergio Serrada-Tejeda, Cristina García-Bravo, Marta Pérez-de-Heredia-Torres

**Affiliations:** 1Department of Physical Therapy, Occupational Therapy, Rehabilitation and Physical Medicine, King Juan Carlos University, Avenida de Atenas s/n., Alcorcon, 28922 Madrid, Spain; 2Department of General Medicine, A Mariña and Monforte de Lemos Health Area, 27002 Lugo, Spain

**Keywords:** Long COVID, headaches, dizziness, disability

## Abstract

Background: Currently, about 15% of coronavirus disease-19 (COVID-19) patients are affected by Long COVID worldwide; however, this condition has not yet been sufficiently studied. The aim of this study was to identify the impact of symptom persistence as well as clinical and socio-demographic variables in a cohort of people with Long COVID. Methods: We conducted a descriptive cross-sectional study of a sample of adult patients from different Spanish regions presenting with Long COVID. Data collection was conducted between April and July 2021. Functional status and dependency were assessed. Results: A multivariate linear regression was performed, and the model was statistically significant (F (7; 114) = 8.79; *p* < 0.001), according to the overall ALDQ score. The variables with a statistically significant effect on the degree of dependence were age (*p* = 0.014), time since diagnosis (*p* = 0.02), headaches (*p* = 0.031), and dizziness (*p* = 0.039). Functional status post-COVID showed a positive and significant relationship with the percentage of dependence (*p* < 0.001). Conclusions: People affected by Long COVID showed moderate dependency status and limitations in functionality. Those with neurological symptoms, such as dizziness and headaches, as well as older age, showed a higher degree of dependency. Improvements in dependency status occurred with increasing time since diagnosis.

## 1. Introduction

The so-called severe acute respiratory syndrome coronavirus 2 (SARS-CoV-2) gave rise to an epidemic disease known as coronavirus disease-19 (COVID-19). The symptoms caused by this virus range from general symptoms such as fever, nausea and vomiting, shortness of breath, cough, muscle aches, and fatigue [1,2] to neurological symptoms such as headaches, dizziness, vertigo, mental fog [3], and, in the worst cases, death. The term Long COVID is a condition in which people present symptoms despite having passed the acute phase of the disease that persist much longer than expected [4]. Specifically, Greenhalgh et al. [5] defined “post-acute COVID-19” as the presence of symptoms that extends beyond three weeks and “chronic COVID-19” as symptoms that last beyond 12 weeks. The continuation of symptoms may be due to the combination of a cytokine storm and entry of the virus into the central nervous system (CNS), which can cause neuroinflammation that may result in prolonged and very frequent neurological symptoms. These symptoms severely decrease the quality of life of those affected [6]. Regarding the percentage of patients affected, Stavem et al. [7] demonstrated that after the onset of COVID-19, there was a low percentage of patients with symptoms lasting longer than six months. However, globally, this condition affects around 10% of patients, with an estimated health burden across all age groups of 30% [8], leading to a large economic impact [9]. Notably, more than 200 different symptoms have been described, among which fatigue and dyspnoea are the most frequent symptoms. In addition, headaches are present in 84% of cases, and dizziness in 69.4% of cases, both of which have been found to be very disabling [10]. Regarding socio-demographic characteristics, Long COVID can affect people of any age, sex, and condition, but in more than 50% of the cases, the affected are between 36 and 50 years old (average age 43). Of these, 70–80% are women, and most of them do not have comorbidities [10,11].

The medium- and long-term effects of COVID-19 infections on patients’ general health, well-being, physical function, and ability to return to work has not yet been studied in depth [12]. A preliminary study by Belli et al. [13] demonstrated impairments in physical functioning and performance of activities of daily living (ADLs) upon hospital discharge. Carfi et al. [12] reported a persistence of symptoms and reduced quality of life beyond sixty days after symptom onset, and Taboada et al. [14] demonstrated that 47.5% of patients with COVID-19 had decreased functional status six months after hospitalisation. Most authors have focused on hospitalised patients [13]; however, it has been shown that the COVID-19 infection results in at least two clinically distinct types of post-COVID syndrome. Specifically, the two types are post-COVID syndrome in hospitalised patients [15] or a mild-to-moderate infection that does not require hospitalisation. The latter type of patients, due to the preventive measure of social isolation, have received limited healthcare resources and less follow-up, which puts them at a disadvantage compared to other pathologies known to date [16]. It is, therefore, necessary to investigate the limitations these patients present to carry out an appropriate intervention and rehabilitation to improve their quality of life and autonomy [17].

The aim of this study was to identify the impact of Long COVID on ADLs and to determine the effect of clinical and socio-demographic variables on the severity of functional status post-COVID, as well as on the degree of dependency and limitations affecting the performance of activities.

## 2. Materials and Methods

### 2.1. Design

This was a descriptive cross-sectional study involving a sample of Spanish adult patients presenting with persistent COVID-19 symptoms lasting three months or longer. This study was approved by the ethics committee of university nº 1701202102121. Data collection, processing, and transfer were completed in accordance with the Declaration of Helsinki [18] and current Spanish regulations on personal data protection. Furthermore, each participant signed an informed consent form. 

### 2.2. Participants

Data collection was conducted between April and July 2021. The survey method adopted used videoconference with the affected persons. Using only volunteering participants who fulfilled the criteria, patients were selected by simple random sampling using the 2022 Quick-Calcs GraphPad software system (GraphPad Software, LLC, San Diego, CA, USA). The selection of the sample was determined by the Sociedad Española de Médicos Generales y de Familia (SEMG), based on previous national and international studies. The inclusion criteria consisted of people who were aged between 30 and 50 years, were diagnosed of COVID-19 disease by PCR and/or positive serology without hospitalisation, had persistent symptomatology for three months or more due to COVID-19 determined by medical diagnosis, had adequate communication skills to collect clinical data, and had no previous pathologies, (which included not suffering from headaches and migraines previously). The exclusion criteria included not having received rehabilitation treatment for COVID-19, not having the necessary technology to carry out the interview, and the subject’s failure to accept and sign the informed consent form.

### 2.3. Procedure

The project was agreed upon with the SEMG, representatives of the “Long Covid Autonomous Communities Together Spain (ACTS)” collective; subsequently, Long Covid ACTS conveyed the study information to the regional collectives of each community, and, in turn, they disseminated the information to each of the affected individuals who voluntarily showed their interest in participating in the study. The form included contact details, a COVID-19 diagnostic test positive, time since diagnosis, vaccination against SARS-CoV-2, symptomatology, description of symptomatology (frequency, intensity of pain, duration), lifestyle, and acceptance of the study. Once the participant had completed the form and given their informed consent, the researcher contacted the participant via videoconference. During the interview, the researcher administered the corresponding assessment scales of Activities of Daily Living Questionnaire (ADLQ) and Post-COVID-19 Functional Status Scale (PCFS). These scales were administered considering the current situation, and the same questions were also administered with reference to the pre-disease situation. Subsequently, the data were stored in a pseudonymised digital booklet with a code.

### 2.4. Measures

The Activities of Daily Living Questionnaire (ADLQ) [19] is an assessment tool that can be self-administered by the patient or administered by their caregiver which measures patient’s functional ability in relation to different ADLs. Function is measured in six areas: self-care, home care and management, employment and leisure, shopping and money management, transportation, and communication. This scale is composed of 28 items that are scored from 0 (no problem) to 4 (can no longer perform the activity). Total scores and subscales were expressed as a percentage to indicate the degree of dependency. The degree of impairment is classified as “severe” (>66%), “moderate” (34–66%) or “none to mild” (0–33%). Satisfactory psychometric properties of the ADLQ have been demonstrated with strong internal consistency (Cronbach’s *p* = 0.88) and concurrent validity (significant correlations with CDR and FAQ, both *p* < 0.001) [20].

The Post-COVID-19 Functional Status Scale (PCFS) [21] is an assessment tool used to identify alterations in post-COVID-19 functional status and its evolution over time. It consists of six ordinal categories reflecting conditions of increasing severity. It covers the full range of functional areas limitations in usual tasks/activities, both at home and in the workplace, as well as changes in lifestyle. This scale includes five grades of increasing severity, from 0 to 4. Grade 4 is the most severe and D is recorded for “death”. This scale represents a new method of patient assessment in the post-COVID-19 phase that has already been used in this type of patients and has previously been shown to have adequate psychometric properties in terms of reliability and construct validity, with translations and cultural adaptations available for different countries [22].

### 2.5. Statistical Analysis

Regarding the qualitative variables, the number of cases present in each category and the corresponding percentages were calculated, and for quantitative variables, the mean and standard deviation were calculated. Correlations between variables were studied using Pearson’s correlation coefficient. To determine the possible effect of demographic, clinical, and scale variables, multivariate linear regression models were performed for the global score and for the dimensions. Statistical analysis was performed with SPSS 27.0 (SPSS Inc., Chicago, IL, USA) for Windows (Copyright© 2022 IBM SPSS Corp.). Statistically significant differences were those with a *p*-value less than 0.05.

## 3. Results

The final study sample consisted of 122 patients from 35 Spanish territories aged between 30 and 50 years, with a mean age of 43.5 years (SD = 5.8), of whom 77.9% (n = 95) were women and 22.1% (n = 27) were men. The mean time since COVID-19 diagnosis was ten months. Persistent symptomatology reported by participants included headaches (71%), dizziness (59%), and paraesthesia affecting the extremities (65%). All participants described a fluctuating and daily persistence of symptoms with mild-to-moderate intensity in the case of headaches. Table 1 shows the descriptive summary of the clinical variables analysed in this study.

Table 2 shows the measurement and correlations of the results for the dependence and functional status assessments. Regarding the latter two scales, the dependency percentage was moderate (ADLQ = 31.37 ± 9.65), and the participants showed a post-COVID functional status with a mild-to-moderate degree of impairment (PCFS = 2.87 ± 0.82). The dependency status scores for each of the dimensions were based on basic self-care activities (self-care = 4.12 ± 2.19), the ability to perform household maintenance activities (home care and management = 8.44 ± 3.09), work performance and leisure (employment and recreation = 5.93 ± 2.16), the ability to make financial transactions and purchases (purchases and money management = 2.31.44 ± 1.56), the ability to manage and handle oneself in one’s environment (travel = 5.57 ± 2.29), and expression through oral and written comprehension (communication = 5.34 ± 1.95). All correlations were positive; therefore, both the ADLQ total score and its dimensions showed a statistically significant and positive relationship with the PCFS.

To determine the possible influence of clinical variables and the PCFS on the ADLQ and its dimensions, a multiple linear regression model adjusted for gender, age, and time of evolution was performed, the results of which are shown in Table 3.

For the total score, the model was statistically significant F (7;114) = 8.79; *p* < 0.001), explaining 31.1% of the variability of the score. Of the demographic variables, age had a statistically significant effect (*p* = 0.014) meaning that increasing age of the participants was associated with a higher state of dependence, as measured by the ADLQ. Sex had no significant effect on functional status; therefore, no differences by gender were shown. As for the clinical variables, the time since diagnosis was significant, although in this case, with a negative effect (*p* = 0.02). The results showed significant differences in headaches (*p* = 0.031) and dizziness (*p* = 0.039). In relation to the PCFS scale, which determined limitations in functional status, a positive and significant relationship was found with the ADLQ scale (*p* < 0.001), and, therefore, a greater degree of deterioration in functional status post-COVID was related to a greater degree of dependence (Table 3).

The impact of clinical variables on the dimensions of self-care activities, home care and management, and employment and recreation were also analysed. Table 4 shows the results of the models conducted to test the effect of demographic and clinical variables.

Regarding the self-care activities, age (*p* = 0.012) and dizziness (*p* = 0.034) showed a significant effect. Concerning the ability to perform care and household management activities, the variables that showed a significant effect were headaches (*p* = 0.025) and dizziness (*p* = 0.034). In the employment and recreation dimension, age (*p* = 0.02), time of evolution (*p* = 0.04), dizziness (*p* = 0.039), and paraesthesia of extremities (*p* = 0.021) were significant. Finally, the relationship of functional post-COVID status measured by PCFS was significant (*p* < 0.01) in all three dimensions, showing a direct relationship on the performance of activities in these dimensions (Table 4).

Regarding the effect of the variables on the dimensions of shopping and money, travelling, and communication (Table 5), time of evolution was significant (*p* = 0.08) for the management of purchases and money management. In the ability to travel and move around, the variables of time of evolution (*p* = 0.035) and dizziness (*p* = 0.012) showed a significant effect. In the ability to communicate and understand, the variables with a significant effect were headaches (*p* = 0.044) and dizziness (*p* = 0.045). The direct effect of functional status post-COVID was significant for all three dimensions (*p* < 0.001).

## 4. Discussion

To our knowledge, this is the first study to analyse functional status and dependency post-COVID and the effect of neurological symptomatology on the severity of people with Long COVID. Different authors such as Shah et al. and Cares et al. [23,24] have pointed out that symptoms such as fatigue and dyspnoea are frequent in patients with Long COVID and that these symptoms have a negative impact on the functionality and quality of life of those affected. However, the evidence available on the impact of other types of symptoms, such as dizziness and headaches in patients affected by Long COVID, is greatly limited, although previous studies have shown that they are also very frequent in the long term and that they seriously affect quality of life [25,26].

Straburzyński M et al. [27] analysed the causes of headaches, and their results pointed to an innate immune response with important clinical consequences, which could explain the impact on the daily life of those affected. In line with previous results in the literature, there were significant differences in headaches (*p* = 0.031) based on the severity of the dependency status of participants. In addition, patients reported chronic daily headaches of mild-to-moderate intensity that varied and worsened with activity.

Concretely, people affected by Long COVID who suffered from headaches showed a higher percentage of disability and, specifically and significantly, showed greater limitations in the dimensions that include the performance of activities such as meal preparation, cleaning, laundry, and household repairs (household care and management: *p* = 0.025) and difficulties affecting the ability to express themselves and understand conversations, also impairing reading and writing (communication: *p* = 0.044). These results support the findings of previous similar studies, such as that of Garcia et al. [28] who focused on analysing headaches as a disabling, lingering symptom in patients affected by COVID-19 and concluded that it causes personal suffering, impaired quality of life, and a considerable economic burden. A cross-sectional study by Stadio et al. [25] analysed the frequency of headaches beyond six months after diagnosis with COVID-19 and their strong association with mental fog. In this regard, Hansen et al. [29] concluded that headaches directly affected the individual’s ability to attend to and concentrate on activities, which may concur with the results shown in this study; however, neither of these authors focused their analysis on non-hospitalised Long COVID sufferers.

Along these lines, our results showed a significant effect of dizziness (*p* = 0.039) on the severity of disability and dependency status, as well as limitations in the dimensions encompassing the performance of dressing, grooming, and personal grooming activities (self-care: *p* = 0.034), household management and care (*p* = 0.034), socio-occupational difficulties (employment and recreation dimension: *p* = 0.039), driving and transport management, travel (travel: *p* = 0.012) and communication (*p* = 0.045), all of which showed significant differences. Saniasiaya et al. [25] analysed the frequency and impact of persistent dizziness due to COVID-19, showing the need for referral and comprehensive research to tailor appropriate rehabilitation therapy. In this regard, previous studies in the literature have shown that dizziness often leads to considerable changes in the lifestyle of those affected, which may include reduced participation in or avoidance of ADLs, reduced health-related quality of life and reduced well-being [30,31].

Regarding the functionality and independence of this population in ADLs, the available evidence remains limited [11]. Our results indicate that non-hospitalised Long COVID patients have a moderate dependency rate (ADLQ = 31.37 ± 9.65) and a post-COVID functional status of grade PCFS = 2.87 ± 0.82, which translates to a mild-to-moderate degree of impairment; specifically, 77% of the participants obtained a grade 3. In agreement with our results, similar studies, such as that of Fernández et al. [32], also showed moderate-to-severe ADL limitations in their results; however, they focused on hospitalised patients and did not determine the functional status post-COVID. Moreover, only one measure was used for the assessment of limitations. In this sense, previous similar studies [12,24] with hospitalised patients with post-COVID syndrome showed limitations in ADL independence; however, unlike our study, they used the Barthel Index as a measure to determine the degree of independence in ADL. This assessment tool is widely recognised and standardised. However, it may not be sufficient, as it only measures independence in basic activities, failing to consider other, not purely physical deficits that may affect performance and functionality [33]. Analysing these deficits can be crucial in directing rehabilitation efforts towards deficits in activities affected by low self-efficacy [34], as demonstrated by our results.

Moreover, regarding the assessment of functionality, it is necessary to use a specific tool for Long COVID, such as the PCFS. This research provides relevant data, as we show a significant correlation in the results between functional status post-COVID and its direct relationship with the degree of disability and dependence (*p* < 0.001). Therefore, we can hypothesise that it is a valid and reliable tool that requires future studies to acquire greater relevance [35] and to support its clinical application [22].

Currently, to our knowledge, no previous published studies have analysed the time since diagnosis and the long-term post-COVID functional status of non-hospitalised affected individuals; most authors have focused on accounting for signs and symptoms present at the time of the last assessment [36]. Our results provide relevant findings showing significant results in this clinical variable. The negative effect observed (time of evolution: *p* = 0.02) shows that with greater time since the diagnosis of COVID-19, improvements were found in their status of dependency and disability. Specifically, and significantly, improvements were found in the ability for people affected by Long COVID to manage purchases and money (purchases and money dimension: *p* = 0.008), the ability move around and travel (travel dimension: *p* = 0.035), and social interaction and reincorporation to work (employment and recreation dimension: *p* = 0.004). In this sense, most studies, such as Carfi et al. [11] and Di Stadio et al. [25], analysed the symptoms of people affected only six months after infection. Other more recent studies, such as a study by Fernandez et al. [37], conducted with post-COVID patients who had been hospitalised in the acute state of the disease, concluded that two years after diagnosis they had less symptomatology; however, they focused on the description of present symptoms only, without determining the functional status of the affected persons and their socio-occupational reincorporation. Norrefalk et al. [38], in their cross-sectional study, recruited a sample of patients with post-COVID syndrome who initially had a mild infection, and their results showed limitations in bodily function and activity. However, it was not related to time since diagnosis, leaving the analysis of sociodemographic variables for future reviews. Along these lines, a longitudinal cohort study by Huang et al. [38] showed that for most COVID-19 sufferers who had been hospitalised, their health status was still lower than that of the control population one year later. However, the authors only focused on the sequelae of acute COVID-19. The results of this study provide results that may be relevant, as they show that when the time from diagnosis is longer, affected people show an improvement in their status of dependency.

Our data show that age had a statistically significant effect (*p* = 0.014), meaning that the closer the patient’s age was to 50 years, the greater the association between a greater dependency status and greater restrictions and difficulty in the performance of basic self-care ADLs and socio-occupational reincorporation. Similar studies have already shown that this sociodemographic variable is a risk factor that aggravates acute SARS-CoV-2 disease due to different causes [38,39]. A meta-analysis by Barek et al. [40] concluded that the prognosis and severity of COVID-19 disease could be affected in patients aged 50 years or older. Along the same vein, Cabrera et al. [36], in their peer-reviewed systematic review, concluded that older age was a risk factor that was potentially associated with developing Long COVID. Mohamed et al. [35] analysed the functional status of post-COVID sequelae in 400 patients, the results of which showed that varying degrees of functional impairment were affected by age. However, none of these authors focused on non-hospitalised patients, and the most affected activities were not comprehensively analysed. 

Our study suffers from several limitations. First, the sample size, only non-hospitalised individuals without comorbidities and aged 30–50-years old were included; therefore, we cannot extrapolate the current results. For the cross-sectional design, the results are based on patient-reported information only; therefore, special caution should be taken to avoid possible measurement bias. In addition, retrospective information was required for the pre-disease questions, with a possible recall bias [41]. We cannot conclude that the headache and dizziness could be part of a larger complex of clinical symptoms that together lead to limitation of daily activities. However, despite these limitations, the present study has enabled an analysis of the degree of dependency and functional status, as well as the impact of neurological symptoms, age, sex, and time of evolution in non-hospitalised patients affected by Long COVID. Therefore, these data can guide in the follow-up of patients and assist in a rehabilitation process appropriate to the needs of these people.

## 5. Conclusions

The results of this study indicate that people affected by Long COVID display a moderate degree of dependence, a mild-to-moderate degree of functional status, and limitations in the performance of their ADLs. In addition, those with neurological symptoms, such as dizziness and headaches, as well as older age, showed a higher degree of dependency than those without. There were improvements in the dependency status of participants with longer time since diagnosis, and there was a direct relationship between the degree of post-COVID functional status with the percentage of dependency of those affected. Future longitudinal studies, with a larger sample in different populations, should be conducted to verify our findings and research the evolution and long-term follow-up in non-hospitalised patients diagnosed with Long COVID-19.

## Figures and Tables

**Table 1 jcm-11-05904-t001:** Descriptive clinical variables.

	Mean (SD)	*n* (%)
**Time evolution**	10.88 (3.33)	
**Headaches**		
No		35 (29.3)
Yes		87 (70.7)
**Dizziness**		
No		50 (41.5)
Yes		72 (58.5)
**Par** **aesthesia/tingling of extremities**		
No		42 (35.0)
Yes		80 (65.0)

**Table 2 jcm-11-05904-t002:** Means (SD) and correlations of the scales.

	Mean (SD)	1				5			8
**ADLQ Total**	31.73 (9.65)	1							
**2. Self-care activities**	4.12 (2.19)	0.75 *	1						
**3. Home care and management**	8.44 (3.09)	0.76 *	0.49 *	1					
**4. Employment and recreation**	5.93 (2.16)	0.73 *	0.49 *	0.42 *	1				
**5. Shopping and money**	2.31 (1.56)	0.61 *	0.27 *	0.32 *	0.43 *	1			
**6. Travel**	5.57 (2.29)	0.76 *	0.45 *	0.51 *	0.45 *	0.41 *	1		
**7. Communication**	5.34 (1.95)	0.70 *	0.55 *	0.32 *	0.42 *	0.44 *	0.45 *	1	
**8. PCFS**	2.87 (0.62)	0.51 *	0.45 *	0.36 *	0.42 *	0.27*	0.37 *	0.35 *	1

* *p* < 0.001.

**Table 3 jcm-11-05904-t003:** Effect of demographic and clinical variables on the total ADLQ score.

	B (ET)	*t*	*p*-Value
**Sex (Female vs. Male)**	1.39 (1.84)	0.75	0.452
**Age**	0.32 (0.13)	2.49	**0.014**
**Time of evolution**	−0.56 (0.24)	−2.36	**0.002**
**Headaches (Yes vs. No)**	1.46 (0.67)	2.18	**0.031**
**Dizziness (Yes vs. No)**	1.44 (0.69)	2.07	**0.039**
**Tingling limbs (Yes vs. No)**	1.11 (1.63)	0.68	0.499
**PCFS**	8.23 (1.22)	6.76	**<0.001**

**Table 4 jcm-11-05904-t004:** Effect of demographic and clinical variables on the self-care activities, home care and management, and employment and recreation dimensions of the ADLQ scale.

	Self-Care Activities			Home Care and Management			Employment and Recreation		
	B (ET)	*t*	*p*-Value	B (ET)	*t*	*p*-Value	B (ET)	*t*	*p*-Value
**Sex (Female vs. Male)**	0.38 (0.43)	0.88	0.38	0.10 (0.67)	0.16	0.877	−0.28 (0.43)	−0.65	0.518
**Age**	0.08 (0.03)	2.55	**0.012**	0.06 (0.05)	1.27	0.205	0.10 (0.03)	3.25	**0.002**
**Time evolution**	−0.08 (0.06)	−1.36	0.178	−0.09 (0.09)	−1.01	0.317	−0.11 (0.05)	−2.08	**0.004**
**Headaches (Yes vs. No)**	0.38 (0.39)	0.98	0.328	0.68 (0.30)	2.27	**0.025**	0.01 (0.39)	0.04	0.971
**Dizziness (Yes vs. No)**	0.60 (0.28)	2.14	**0.034**	0.58 (0.27)	2.15	**0.034**	0.46 (0.22)	2.09	**0.039**
**Tingling limbs (Yes vs. No)**	0.46 (0.39)	1.21	0.23	−0.64 (0.60)	−1.07	0.285	0.90 (0.39)	2.35	**0.021**
**PCFS**	1.56 (0.29)	5.42	**<0.001**	1.90 (0.44)	4.29	**<0.001**	1.40 (0.29)	4.87	**<0.001**
***R*^2^ (%)**	25.8			19.7			23.9		
**Model**	*F* (7; 114) = 7.01; *p* < 0.001			*F* (7; 114) = 3.15; *p* = 0.004			*F* (7; 114) = 6.44; *p* < 0.001		

**Table 5 jcm-11-05904-t005:** Effect of demographic and clinical variables on the shopping and money, travel, and communication dimensions of the ADLQ scale.

	Shopping and Money			Travelling			Communication		
	B (ET)	*t*	*p*-Value	B (ET)	*t*	*p*-Value	B (ET)	*t*	*p*-Value
**Sex (Female vs. Male)**	0.75 (0.33)	2.26	0.26	0.38 (0.49)	0.78	0.438	0.05 (0.42)	0.13	0.898
**Age**	0.00 (0.02)	−0.18	0.861	0.05 (0.03)	1.56	0.121	0.04 (0.03)	1.26	0.212
**Time evolution**	−0.12 (0.04)	−2.69	**0.008**	−0.13 (0.06)	−2.13	**0.035**	−0.06 (0.05)	−1.10	0.275
**Headaches (Yes vs. No)**	−0.10 (0.30)	−0.35	0.73	0.34 (0.44)	0.78	0.44	0.53 (0.26)	0.04	**0.044**
**Dizziness (Yes vs. No)**	0.05 (0.28)	0.18	0.857	0.41 (0.16)	2.56	**0.012**	0.70 (0.35)	2.03	**0.045**
**Tingling limbs (Yes vs. No)**	0.12 (0.29)	0.42	0.676	0.23 (0.43)	0.54	0.591	0.02 (0.37)	0.06	0.951
**PCFS**	0.81 (0.22)	3.70	**<0.001**	1.47 (0.32)	4.57	**<0.001**	1.09 (0.27)	3.99	**<0.001**
***R*^2^ (%)**	21.1			19.9			19.3		
**Model**	*F* (7; 114) = 3.88; *p* = 0.001			*F* (7; 114) = 3.94; *p* = 0.001			*F* (7; 114) = 3.95; *p* = 0.001		

## Data Availability

All data are available upon request from the corresponding author.

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
