# Peer review of "Headaches and Dizziness as Disabling, Persistent Symptoms in Patients with Long COVID–A National Multicentre Study"

_jcm, 2022, doi:10.3390/jcm11195904_

Round 1

Reviewer 1 Report

I read your article with an interest. I agree with the authors that most published articles on long COVID described the prevalence of symptoms but not how it impacted patients’ well-being and daily living activities (DLA), which is the strength of this article.

However, I have a few concerns, which I hope the authors address satisfactorily.

1 Despite the inclusion of 35 Spanish territories, only 122 patients were included. The authors claim that this represents national data is slightly misleading. I think this is one of the main limitations of the study. I would appreciate it if the authors included this in their limitations.

2 What was the rationale behind this study's 55-year-old age limit for patient recruitment? Do you think this might have skewed the results?

3 In your study, age was statistically significant for disability. Have the authors considered the patients’ co-morbidities? Have the authors recorded any co-morbidities that might have influenced the patients’ DLA before they developed long COVID? It would be useful if the authors discussed patients’ co-morbidity and how it might have impacted patients’ DLA.

4 Do you believe there is enough evidence so far to suggest that PCFS can be used as a valid tool to assess long COVID-related DLA? 

5 I feel discussion can be concise. Paragraphs 2 and 6 discuss the same points, namely that several long COVID studies elaborated on long COVID symptomology rather than its effects on DLA. I suggest re-writing these paragraphs.

6 This sentence is difficult to read. A in although should be a small letter. I suggest breaking this long sentence into two for better readability. ‘’In this sense, previous similar studies [11,23] demonstrated by our results.’’

Author Response

Response letter manuscript

ID: jcm-1942471

“Headaches and dizziness as disabling persistent symptoms in patients
with Long COVID. A national multicentre study”.

We would like to thank the editor and reviewers for their comments in this review, which have greatly improved the readability of the manuscript. We would like to inform you that we have edited the manuscript according to the very constructive suggestions from the reviewers.

Below, please find a list of revisions and a response to each of the reviewer’s comments. We have highlighted all changes in the manuscript in light blue. We hope that the revisions in the manuscript and our accompanying responses will be enough to make our manuscript suitable for publication in the Journal of Clinical Medicine

We shall look forward to hearing from you at your earliest convenience.

Yours sincerely,

The authors

REVIEWER 1

I read your article with an interest. I agree with the authors that most published articles on long COVID described the prevalence of symptoms but not how it impacted patients’ well-being and daily living activities (DLA), which is the strength of this article.

However, I have a few concerns, which I hope the authors address satisfactorily.

1 Despite the inclusion of 35 Spanish territories, only 122 patients were included. The authors claim that this represents national data is slightly misleading. I think this is one of the main limitations of the study. I would appreciate it if the authors included this in their limitations.

Response: Thank you for your comments. Given your recommendations. We have added a paragraph referring to this aspect in our limitations of study.

Line 307-311: Our study suffers from several limitations. Firstly, the sample size, only no hospital-ized individuals, without comorbidities and aged 30-50-years old were included, there-fore, we cannot extrapolate the current results.  The cross-sectional design, the results are based on patient-reported information only, therefore, special caution should be taken to avoid possible measurement bias.

2 What was the rationale behind this study's 55-year-old age limit for patient recruitment? Do you think this might have skewed the results?

Response: We appreciate your feedback as it helps us to improve the quality of the paper.

Based on previous research, different studies have shown the frequency of non-hospitalised Long COVID in its acute phase, being adults of an average age of 43 years, which is why we have focused on this study. We have added information about it in the introduction and participants’ sections.

Lines: 48-50: About socio-demographic characteristics, Long COVID can affect people of any age, sex, and condition, but more than 50% of the cases are between 36 and 50 years old (average age 43), 70-80% are women and most of them do not have comorbidities [10,11].

Lines: 84-86: The selection of the sample was determined by the Sociedad Española de Medicos Generales y de Familia (SEMG), based on previous national and international studies [10,11]

We agree that age may be a bias that limits the interpretability of the results. However, previous studies on non-hospitalised Long Covid patients, provide socio-demographic data indicating that the average age 43 [10,11]

We have added information about it in the limitations and conclusions’ sections.

Lines: 307-311: Our study suffers from several limitations. Firstly, the sample size, only no hospital-ized individuals, without comorbidities and aged 30-50-years old were included, there-fore, we cannot extrapolate the current results.  The cross-sectional design, the results are based on patient-reported information only, therefore, special caution should be taken to avoid possible measurement bias.

3 In your study, age was statistically significant for disability. Have the authors considered the patients’ co-morbidities? Have the authors recorded any co-morbidities that might have influenced the patients’ DLA before they developed long COVID? It would be useful if the authors discussed patients’ co-morbidity and how it might have impacted patients’ DLA.

Response: Thank you for your comments. In this research only patients without comorbidity were included to analyse the effects long COVID in the healthy population. The selection of the sample was determined by the Sociedad Española de Medicos Generales y de Familia (SEMG), based on previous national and international studies. However, we agree that it may be interesting in future studies to carry out an analysis in other sample population.

Added information in:

Lines 48-50: About socio-demographic characteristics, Long COVID can affect people of any age, sex, and condition, but more than 50% of the cases are between 36 and 50 years old (average age 43), 70-80% are women and most of them do not have comorbidities [10,11]

Lines 88-91: […] with persistent symptomatology due to covid 19 determined by medical diagnosis, for three months or more, adequate communication skills to collect clinical data, and no previous pathologies, (which included not suffering from headaches and migraines previously).

Limitations’ section:

Lines 307-310: Our study suffers from several limitations. Firstly, the sample size, only no hospital-ized individuals, without comorbidities and aged 30-50-years old were included, there-fore, we cannot extrapolate the current results. The cross-sectional design, the results are based on patient-reported information only, therefore, special caution should be taken to avoid possible measurement bias.

4 Do you believe there is enough evidence so far to suggest that PCFS can be used as a valid tool to assess long COVID-related DLA? 

Response: Thanks for your comments. Different post COVID studies have demonstrated the psychometric properties, validity, and reliability of the scale in different countries. We agree with you that more research is needed to bring more validity and trajectory to this assessment instrument. Therefore, other ADLs have been used to correlate the results with other assessments of ADLs with wide validity and validity and to correlate the results with other ADLs.

Lines 128-130: This scale represents a new method of patient assessment in the post-COVID-19 phase that has already been used in this type of patients and has previously been shown to have adequate psychometric properties in terms of reliability and construct validity, with translations and cultural adaptations available for different countries [21].

5 I feel discussion can be concise. Paragraphs 2 and 6 discuss the same points, namely that several long COVID studies elaborated on long COVID symptomology rather than its effects on DLA. I suggest re-writing these paragraphs.

6 This sentence is difficult to read. A in although should be a small letter. I suggest breaking this long sentence into two for better readability. ‘’In this sense, previous similar studies [11,23] demonstrated by our results.’’

Response: Thank you for your comment, the text has been revised it has been changed.

Lines 241-244: In this regard, previous literature has shown that dizziness often leads to considerable changes in the lifestyle of those affected, which may include reduced participation in or avoidance of ADLs, reduced health-related quality of life and reduced well-being [29,30] according to the results of this study.

Response: Thank you very much for your comment. We are change has been made to the sentence structure.

Lines: 253-259: In this sense, previous similar studies [11,23] with hospitalised patients with post-COVID syndrome showed limitations in ADL independence, however, unlike our study, they used the Barthel Index as a measure to determine the degree of independence in ADL. The assesment tool is a widely recognised standardised, it may not be sufficient as it only measures independence in basic activities, failing to consider other, not purely physical deficits that may affect performance and functionality [32], as demonstrated by our results.

Reviewer 2 Report

Congratulations to the authors for this manuscript. It is easy to read, with understandable and simple writing. 

Minnor comments:

-Provide the registration number of the Bioethics Committee in the manuscript in the design section.

- Include practical implications in the discussion section.

Author Response

Response letter manuscript

ID: jcm-1942471

“Headaches and dizziness as disabling persistent symptoms in patients
with Long COVID. A national multicentre study”.

We would like to thank the editor and reviewers for their comments in this review, which have greatly improved the readability of the manuscript. We would like to inform you that we have edited the manuscript according to the very constructive suggestions from the reviewers.

Below, please find a list of revisions and a response to each of the reviewer’s comments. We have highlighted all changes in the manuscript in light blue. We hope that the revisions in the manuscript and our accompanying responses will be enough to make our manuscript suitable for publication in the Journal of Clinical Medicine

We shall look forward to hearing from you at your earliest convenience.

Yours sincerely,

The authors

REVIEWER 2

Congratulations to the authors for this manuscript. It is easy to read, with understandable and simple writing. 

Provide the registration number of the Bioethics Committee in the manuscript in the design section.

Response: Thanks for your comment, we have added the number in the text.

Line 76: This study was approved by the ethics committee of university nº 1701202102121

Include practical implications in the discussion section.

Response: Thanks for your comment. We have added a paragraph referring to this aspect in the discussion.

Lines: 317-318: Therefore, these data can guide in the follow-up of patients and assist in a rehabilitation process appropriate to the needs of these people.

Reviewer 3 Report

The paper deals with the interesting issue of long-term symptoms after COVID-19 infection. This is a topic that is still unknown and the coming years will be critical and crucial for understanding the mechanisms of action of this virus on the human body.

However interesting the paper contains some limitations that should be completed and/or corrected before the paper is accepted for publication:

1. The authors showed that up to 71% of the subjects had headaches. But the term headaches is very general and says nothing about this population. We also do not know the history of these patients prior to COVID-19 - did they suffer from migraine or other primary headaches? What is the current phenotyp of the pain? has there been an increase in the severity and frequency of the headaches prior to the infection? did they develop pain after the infection that might suggest a secondary background of the condition? 

2. have the patients had a head imaging study done at least once? it should be remembered that both headaches and dizziness are very common complaints in the general population often having a different pathological cause 

3. were the headaches related to vaccination against SARS-CoV2? we know that they can appear after vaccination and persist for some time, based on a meta-analysis worth citing: https://pubmed.ncbi.nlm.nih.gov/35361131/

4. the discussion lacks a more detailed analysis of the causes of headaches associated with COVID-19, the following review clarifies the topic and should be cited based on information from it: https://pubmed.ncbi.nlm.nih.gov/35758225/

Author Response

Response letter manuscript

ID: jcm-1942471

“Headaches and dizziness as disabling persistent symptoms in patients
with Long COVID. A national multicentre study”.

We would like to thank the editor and reviewers for their comments in this review, which have greatly improved the readability of the manuscript. We would like to inform you that we have edited the manuscript according to the very constructive suggestions from the reviewers.

Below, please find a list of revisions and a response to each of the reviewer’s comments. We have highlighted all changes in the manuscript in light blue. We hope that the revisions in the manuscript and our accompanying responses will be enough to make our manuscript suitable for publication in the Journal of Clinical Medicine

We shall look forward to hearing from you at your earliest convenience.

Yours sincerely,

The authors

REVIEWER 3

The paper deals with the interesting issue of long-term symptoms after COVID-19 infection. This is a topic that is still unknown, and the coming years will be critical and crucial for understanding the mechanisms of action of this virus on the human body.

However interesting the paper contains some limitations that should be completed and/or corrected before the paper is accepted for publication:

1. The authors showed that up to 71% of the subjects had headaches. But the term headaches is very general and says nothing about this population. We also do not know the history of these patients prior to COVID-19 - did they suffer from migraine or other primary headaches? What is the current phenotyp of the pain? has there been an increase in the severity and frequency of the headaches prior to the infection? did they develop pain after the infection that might suggest a secondary background of the condition? 

Response: Thanks for your comment, the questions you raise are very interesting.

The headaches in the sample appeared after covid-19 diagnosis and continued at the time of the assessment. Prior to covid-19 infection, they did not suffer from any type of regular headache, no migraine sufferers that prevented them from performing their ADLs and had no previous co-morbidity. We do not have sufficient data to determine the phenotype of the headache, but we do know that the frequency and mild-moderate intensity of headaches was present in their daily lives and limited their ability to perform their ADLs as reported by the patients.

Added information in:

Participants’ section

Lines 88-91: […] with persistent symptomatology due to covid 19 determined by medical diagnosis, for three months or more, adequate communication skills to collect clinical data, and no previous pathologies, (which included not suffering from headaches and migraines previously).  

Lines 99-104: […] vaccination against SARS-CoV2, symptomatology, description of symptomatology (frequency, intensity of pain, duration), lifestyle and acceptance of the study. Once the participant had completed the form and accepted the informed consent, the researcher contacted the participant via videoconference.

Results’ section

Lines 145-147: All participants described a fluctuating and daily persistence of symptoms with mild-moderate intensity in the case of headache.

2.have the patients had a head imaging study done at least once? it should be remembered that both headaches and dizziness are very common complaints in the general population often having a different pathological cause 

Response: Thanks for your comment. the patients were evaluated by their family doctor in their geographical area and referred to specialists including neurologists and other professionals who determined with diagnostic criteria in their medical report that there were no other pathological causes.

Lines 88-91: with persistent symptomatology due to covid 19 determined by medical diagnosis, for three months or more, adequate communication skills to collect clinical data, and no previous pathologies, (which included not suffering from headaches and migraines previously).  

3. were the headaches related to vaccination against SARS-CoV2? we know that they can appear after vaccination and persist for some time, based on a meta-analysis worth citing: https://pubmed.ncbi.nlm.nih.gov/35361131/

Response: Thank you for your suggestions and comments.

The meta-analysis it suggests is very interesting. We know that subsequent studies have suggested effects of vaccination on a percentage of patients who had Long COVID.

However, our data collection was from April to July 2021 in Spain there was still a high percentage of the population that had not been vaccinated because they were not yet vaccinated according to their age group or because they had had acute covid less than six months ago. Our participants had not received the vaccination schedule at the time of the evaluation.

We have added an information referring to this aspect in the participants.

Lines 100-102: vaccination against SARS-CoV2, symptomatology, description of symptomatology (frequency, intensity of pain, duration), lifestyle and acceptance of the study.

4. the discussion lacks a more detailed analysis of the causes of headaches associated with COVID-19, the following review clarifies the topic and should be cited based on information from it: https://pubmed.ncbi.nlm.nih.gov/35758225/

Response: Thanks for your comment and suggest this interesting meta-analysis We have added a paragraph referring to this aspect in the discussion.

Lines 212-217: Straburzyński M et al. [27] analysed the causes of headaches and their results pointed to an innate immune response with important clinical consequences, which could explain the impact on the daily life of those affected. In line with previous literature results showed significant differences in headaches (p=0.031), based on the severity of the de-pendency status of participants. In addition, patients reported chronic daily headaches of mild-moderate intensity that varied and worsened with activity.

Round 2

Reviewer 1 Report

Dear Authors, Thank you for incorporating changes I have suggested. The manuscript now reads better. 

Best Wishes,

Reviewer 3 Report

The authors have addressed my comments and included them in the revised version of the manuscript.